Accepted at the ICLR 2024 Workshop on AI4Differential Equations In Science

# A Multi-Grained Symmetric Differential Equation Model for Learning Protein-Ligand Binding Dynamics

**Shengchao Liu**[*]
UC Berkeley & Caltech
*shengchao.liu@berkeley.edu*

**Weitao Du**[*]
University of CAS
*duweitao@amss.ac.cn*

**Yanjing Li**
CMU
*yanjing2@andrew.cmu.edu*

**Zhuoxinran Li**
University of Toronto
*zhuoxinran.li@mail.utoronto.ca*

**Vignesh Bhethanabotla**
Caltech
*vbhethan@caltech.edu*

**Nakul Rampal**
UC Berkeley
*nakul.rampal@berkeley.edu*

**Omar Yaghi**
UC Berkeley
*yaghi@berkeley.edu*

**Christian Borgs**
UC Berkeley
*borgs@berkeley.edu*

**Anima Anandkumar**
Caltech
*anima@caltech.edu*

**Hongyu Guo**[†]
National Research Council Canada
*Hongyu.Guo@uottawa.ca*

**Jennifer Chayes**[†]
UC Berkeley
*jchayes@berkeley.edu*

## Abstract

In drug discovery, molecular dynamics (MD) simulation for protein-ligand binding provides a powerful tool for predicting binding affinities, estimating transport properties, and exploring pocket sites. There has been a long history of improving the efficiency of MD simulations through better numerical methods and, more recently, by augmenting them with machine learning (ML) methods. Yet, challenges remain, such as accurate modeling of extended-timescale simulations. To address this issue, we propose NeuralMD, the first ML surrogate that can facilitate numerical MD and provide accurate simulations of protein-ligand binding dynamics. We propose a principled approach that incorporates a novel physics-informed multi-grained group symmetric framework. Specifically, we propose (1) a BindingNet model that satisfies group symmetry using vector frames and captures the multi-level protein-ligand interactions, and (2) an augmented neural ordinary differential equation solver that learns the trajectory under Newtonian mechanics. For the experiment, we design ten single-trajectory and three multi-trajectory binding simulation tasks. We show the efficiency and effectiveness of NeuralMD, with a 2000× speedup over standard numerical MD simulation and outperforming all other ML approaches by up to ˜80% under the stability metric. We further qualitatively show that NeuralMD reaches more stable binding predictions.

## 1 Introduction

The simulation of protein-ligand binding dynamics is one of the fundamental tasks in drug discovery (Kairys et al., 2019; Yang et al., 2020; Volkov et al., 2022). Such simulations of binding dynamics are a key component of the drug discovery pipeline to select, refine, and tailor the chemical structures of potential drugs to enhance their efficacy and specificity. To simulate the protein-ligand binding dynamics, *numerical molecular dynamics (MD)* methods have been extensively developed. However, the numerical MD methods are computationally expensive due to the expensive force calculations on individual atoms in a large protein-ligand system.

To alleviate this issue, *machine learning (ML)* surrogates have been proposed to either augment or replace numerical MD methods to estimate the MD trajectories. However, all prior ML approaches for MD are limited to single-system dynamics (*e.g.*, small molecules or proteins) and not protein-ligand binding dynamics. A primary reason is the lack of large-scale datasets. The first large-scale dataset with binding dynamics was released in May 2023 (Siebenmorgen et al., 2023), and to our knowledge, we are now the first to explore it in this paper. Further, prior ML-based MD approaches limit to studying the MD dynamics on a small time interval (1e-15 seconds), while simulation on a longer time interval (*e.g.*, 1e-9 seconds) is needed for specific tasks, such as detecting the transient and cryptic states (Vajda et al., 2018) in binding dynamics. However, such longer-time MD simulations are challenging due to the catastrophic buildup of errors over longer rollouts (Fu et al., 2022a).

Another critical aspect that needs to be integrated into ML-based modeling is the group symmetry present in protein-ligand geometry. Specifically, the geometric function should be equivariant to rotation and translation (*i.e.*, SE(3)-equivariance). The principled approach to satisfy equivariance is to use vector frames, which have been previously explored for single molecules (Jumper et al., 2021), but not yet for the protein-ligand binding complexes. The vector frame basis achieves SE(3)-equivariance by projecting vectors (*e.g.*, positions and accelerations) into the vector frame basis, and such a projection can maintain the equivariant property with efficient calculations (Liu et al., 2023).

**Our Approach, NeuralMD.** We propose NeuralMD, a multi-grained physics-informed approach designed to handle extended-timestep MD simulations for protein-ligand binding dynamics. Our multi-grained method explicitly decomposes the complexes into three granularities: the atoms in ligands, the backbone structures in proteins, and the residue-atom pairs in the complex, to obtain a scalable approach for modeling a large system. We achieve **group symmetry** in BindingNet through the incorporation of vector frames, and include three levels of vector frame bases for multi-grained modeling, from the atom and backbone level to the residue level for binding interactions.Further, our ML approach NeuralMD preserves the **Newtonian mechanics**. In MD, the movement of atoms is determined by Newton's second law, $F = m \cdot a$, where $F$ is the force, $m$ is the mass, and $a$ is the acceleration of each atom. By integrating acceleration and velocity w.r.t. time, we can obtain the velocities and positions, respectively. Thus in NeuralMD, we formulate the trajectory simulation as a second-order ordinary differential equation (ODE) or second-order stochastic differential equation (SDE) problem. Specifically, we augment derivative space by concurrently calculating the accelerations and velocities, allowing simultaneous integration of velocities and positions.

## 2 PRELIMINIARIES

**Ligands.** In this work, we only consider binding complexes with small molecules as ligands. Small molecules can be treated as sets of atoms in the 3D Euclidean space, $\{f^{(l)}, \boldsymbol{x}^{(l)}\}$, where $f^{(l)}$ and $\boldsymbol{x}^{(l)}$ represent the atomic numbers and 3D Euclidean coordinates for atoms in each ligand, respectively.

**Proteins.** Proteins are essentially chains of amino acids or residues, where there are 20 natural amino acids. Noticeably, amino acids are made up of three components: a basic amino group (-NH$_2$), an acidic carboxyl group (-COOH), and an organic R group (or side chain) that is unique to each amino acid. Additionally, the carbon that connects all three groups is called C$_\alpha$. (We refer to this Wiki page for more details.) In this work, due to the large volume of atoms in proteins, we will use coarse-grained modelings on proteins and binding complexes. With this regard, the **backbone-level** data structure for each protein is $\{f^{(p)}, \{\boldsymbol{x}_N^{(p)}, \boldsymbol{x}_{C_\alpha}^{(p)}, \boldsymbol{x}_C^{(p)}\}\}$, for the residue type and the coordinates of $N - C_\alpha - C$ in each residue, respectively. (We may ignore the superscript in the coordinates of backbone atoms for brevity since such backbone structures are unique for residues in proteins) In addition to the backbone level, for a coarser-grained data structure, we further consider **residue-level** modeling for binding interactions, $\{f^{(p)}, \boldsymbol{x}^{(p)}\}$, where the coordinate of $C_\alpha$ is treated as the residue-level coordinate, *i.e.*, $\boldsymbol{x}^{(p)} \boldsymbol{x}_{C_\alpha}^{(p)}$.

**Molecular Dynamics Simulations.** Generally, molecular dynamics (MD) describes how each atom in a molecular system moves over time, following Newton's second law of motion:

$$F = m \cdot a = m \cdot \frac{d^2 \boldsymbol{x}}{dt^2}, \tag{1}$$

where $F$ is the force, $m$ is the mass, $a$ is the acceleration, $\boldsymbol{x}$ is the position, and $t$ is the time. Then, an MD simulation will take Newtonian dynamics to get the trajectories. The Numerical molecular

dynamics (MD) methods fall into two categories: Newtonian and Langevin dynamics. Newtonian dynamics is ideal for systems with minimal thermal effects or when deterministic trajectories are needed. Conversely, Langevin dynamics is preferred for systems where thermal effects are significant, particularly at finite temperatures. In this study, we introduce two versions: an ordinary differential equation (ODE) solver for Newtonian dynamics and a stochastic differential equation (SDE) solver for Langevin dynamics.

**Problem Setting: Protein-Ligand Binding Dynamics Simulation.** In this work, we focus on simulating the protein-ligand binding dynamics in the semi-flexible setting Salmaso & Moro (2018), *i.e.*, proteins with rigid structures and ligands with flexible movements (Siebenmorgen et al., 2023). Thus, the problem is formulated as follows: suppose we have a rigid protein structure $\{f^{(p)}, \{\boldsymbol{x}_N^{(p)}, \boldsymbol{x}_{C_\alpha}^{(p)}, \boldsymbol{x}_C^{(p)}\}\}$ and a ligand with its initial structure and velocity, $\{f^{(l)}, \boldsymbol{x}_0^{(l)}, \boldsymbol{v}_0^{(l)}\}$. We want to predict the trajectories of ligands following the Newtonian dynamics, *i.e.*, the movement of $\{\boldsymbol{x}_t^{(l)}, ...\}$ over time. We also want to clarify two critical points about this problem setting. (1) We consider trajectory prediction, *i.e.*, positions as labels, and no explicit energy and force labels are considered. ML methods for energy prediction followed with numerical ODE/SDE solver may require smaller timescales (around 1e-15 seconds), while trajectory prediction, which directly predicts the positions, is agnostic to the magnitude of timescales. This is appealing for datasets like MISATO with larger timescales (1e-9 seconds). (2) Each trajectory is composed of a series of geometries of ligands, and such geometries are called **snapshots**.

## 3 METHOD: BINDINGNET AND NEURALMD

In this section, we introduce BindingNet, a multi-grained SE(3)-equivariant geometric model for protein-ligand binding. The input of BindingNet is the geometry of the rigid protein and the ligand at time $t$, while the output is the force on each atom in the ligand. Our architecture is $SE(3)$ equivariant by extending the equivairant frames for small molecules in Du et al. (2022) to 1. Atom level: $\mathcal{F}_{\text{ligand}}$; 2. Protein level: $\mathcal{F}_{\text{protein}}$; 3. $\mathcal{F}_{\text{complex}}$. See appendix D for the formal definition.

**Atom-Level Ligand Modeling.** We first generate the atom embedding using one-hot encoding and then aggregate each atom's embedding, $\boldsymbol{z}^{(l)}$, by aggregating all its neighbor's embedding within the cutoff distance $c$. Then, we obtain the atom's equivariant representation by aggregating its neighborhood's messages as $(\boldsymbol{x}_i^{(l)} - \boldsymbol{x}_j^{(l)}) \cdot \boldsymbol{z}_i^{(l)}$. A subsequent scalarization is carried out based on the atom-level vector frame as $\boldsymbol{h}_{ij}^{(l)} = (\boldsymbol{h}_i^{(l)} \oplus \boldsymbol{h}_j^{(l)}) \cdot \mathcal{F}_{\text{ligand}}$, where $\oplus$ is the concatenation. Finally, it is passed through several equivariant message-passing layers (MPNN). The outputs are atom representation and vector ($\boldsymbol{h}^{(l)}$ and $\text{vec}^{(l)}$), and they are passed to the complex module.

**Backbone-Level Protein Modeling.** For the coarse-grained modeling of proteins, we consider three backbone atoms in each residue. We first obtain the atom embedding on three atom types, and then we obtain each atom's representation $\boldsymbol{z}^{(p)}$ by aggregating its neighbor's representation. Then, we obtain an equivariant atom representation by aggregating the edge information, $(\boldsymbol{x}_i^{(p)} - \boldsymbol{x}_j^{(p)}) \cdot \boldsymbol{z}_i^{(p)}$, within cutoff distance $c$. Following which is the scalarization on the residue frame $\boldsymbol{h}_{ij}^{(p)} = (\boldsymbol{h}_i^{(p)} \oplus \boldsymbol{h}_j^{(p)}) \cdot \mathcal{F}_{\text{protein}}$. Recall that we also have the residue type, and with a type embedding $\tilde{\boldsymbol{z}}^{(p)}$, we can obtain the final residue-level representation using an MPNN layer as $\boldsymbol{h}^{(p)} = \tilde{\boldsymbol{z}}^{(p)} + (\boldsymbol{h}_{N,C_\alpha}^{(p)} + \boldsymbol{h}_{C_\alpha,C}^{(p)})/2$. We leave the **Residue-Level Complex Modeling** in appendix F.

**NODE for molecular dynamics** As clarified in Section 2, molecular dynamics follows Newtonian dynamics, and we solve it as an ordinary differential equation (ODE) problem or a stochastic differential equation (SDE) problem. The BindingNet takes in the molecular system geometry $(\boldsymbol{x}_t^{(l)}, \boldsymbol{x}^{(p)})$ at arbitrary time $t$, and outputs the forces.

To learn the MD trajectory following second-order ODE, we propose the following formulation of the second-order ODE within one integration call:

$$\begin{bmatrix} d\boldsymbol{x}/dt \\ d\boldsymbol{v}/dt \end{bmatrix} = \begin{bmatrix} \boldsymbol{v} \\ F/m \end{bmatrix}, \tag{2}$$

where $F$ is the output forces from BindingNet. This means we augment ODE derivative space by concurrently calculating the accelerations and velocities, allowing simultaneous integration of velocities and positions. Ultimately, following Newtonian mechanics, the coordinates at time $t$ are integrated as:

$$F_\tau^{(l)}\text{-ODE} = \text{BindingNet}(f^{(l)}, \boldsymbol{x}_\tau^{(l)}, f^{(p)}, \boldsymbol{x}_N^{(p)}, \boldsymbol{x}_{C_\alpha}^{(p)}, \boldsymbol{x}_C^{(p)}). \qquad (3)$$

On the other hand, Langevin dynamics introduces a stochastic component for large molecular systems with thermal fluctuations. Concretely, Langevin dynamics is an extension of the standard Newtonian dynamics with the addition of damping and random noise terms:

$$F_\tau^{(l)}\text{-SDE} = \text{BindingNet}(f^{(l)}, \boldsymbol{x}_\tau^{(l)}, f^{(p)}, \boldsymbol{x}_N^{(p)}, \boldsymbol{x}_{C_\alpha}^{(p)}, \boldsymbol{x}_C^{(p)}) - \gamma m \boldsymbol{v} + \sqrt{2m\gamma k_B T} R(t), \qquad (4)$$

$$\boldsymbol{a}_\tau^{(l)} = \frac{F_\tau^{(l)}}{m}, \qquad\qquad \hat{\boldsymbol{v}}_t^{(l)} = \boldsymbol{v}_0^{(l)} + \int_0^t \boldsymbol{a}_\tau^{(l)} d\tau, \qquad \hat{\boldsymbol{x}}_t^{(l)} = \boldsymbol{x}_0^{(l)} + \int_0^t \hat{\boldsymbol{v}}_\tau^{(l)} d\tau. \qquad (5)$$

The objective is the mean absolute error between the predicted coordinates and ground-truth coordinates: $\mathcal{L} = \mathbb{E}_t \big[ |\hat{\boldsymbol{x}}_t^{(l)} - \boldsymbol{x}_t^{(l)}| \big]$. An illustration of NeuralMD pipeline is in appendix Figure 2.

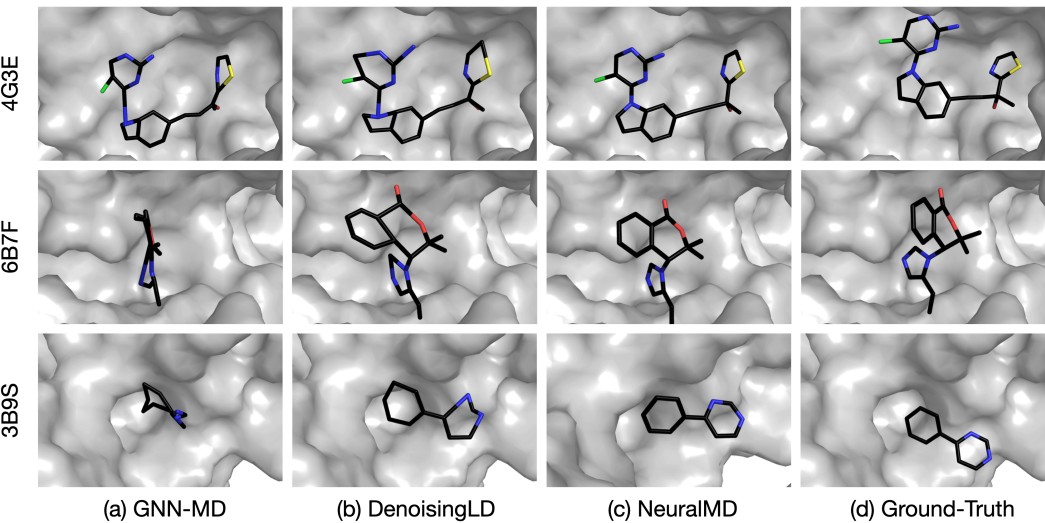

Figure 1: Visualization of last-snapshot binding predictions on three PDB complexes. NeuralMD stays more stable than DenoisingLD, exhibiting a lower degree of torsion with the natural conformations. Other methods collapse heavily, including GNN-MD and VerletMD, where atoms extend beyond the frame for the latter.

## 4 EXPERIMENTS

**Datasets.** We consider MISATO in our work (Siebenmorgen et al., 2023). It is built on 16,972 experimental protein-ligand complexes extracted from the protein data bank (PDB) (Berman et al., 2000). For each protein-ligand complex, the trajectory comprises 100 snapshots in 8 nanoseconds under the fixed temperature and pressure. We want to highlight that MD trajectories allow the analysis of small-range structural fluctuations of the protein-ligand complex. See appendix E for the basic statistics.

**Experiments Settings.** We consider two experiment settings. The first type of experiment is the single-trajectory prediction, where both the training and test data are snapshots from the same trajectory, and they are divided temporally. The second type of experiment is the multi-trajectory prediction, where each data point is the sequence of all the snapshots from one trajectory, and the training and test data correspond to different sets of trajectories.

**Evaluation Metrics.** For both experiment settings, the trajectory recovery is the most straightforward evaluation metric. To evaluate this, we take both the mean absolute error (MAE) and mean squared error (MSE) between the predicted coordinates and ground-truth coordinates. Stability is also an important metric for evaluating the predicted MD trajectory. The intuition is that the prediction on long-time MD trajectory can enter a pathological state (*e.g.*, bond breaking), and stability is

the measure to quantify such observation. It is defined as $\mathbb{P}_{i,j}\big[\big|\|\boldsymbol{x}_i - \boldsymbol{x}_j\| - \boldsymbol{b}_{i,j}\big| > \Delta\big]$, where $\boldsymbol{b}_{i,j}$ is the pair distance at the last snapshot (the most equilibrium state), and we take $\Delta = 0.5$ Å. Another metric considered is frames per second (FPS) on a single Nvidia-V100 GPU card, and it measures the MD efficiency.

**Results on single-trajectory prediction**    We provide the visualization of some examples in Figure 1. Quantitative results are in Table 2 and baselines are introduced in appendix G. The first observation is that the baseline VertletMD has a clear performance gap compared to the other methods. This verifies that using ML models to predict the energy (or force) at each snapshot, and then using a numerical integration algorithm can fail in the long-time simulations (Fu et al., 2022a). Additionally, we can observe that the recovery error of trajectory (MAE and MSE) occasionally fails to offer a discernible distinction among methods (*e.g.*, for protein-ligand complex 3EOV, 1KT1, and 4G3E), though NeuralMD is slightly better. However, the stability (%) can be a distinctive factor in method comparisons, where we observe NeuralMD outperform on all 10 tasks up to ~80%. More detailed analysis is also provided in appendix G.

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

## A   MORE PRELIMINARIES AND RELATED WORK

**Numerical methods for MD** can be classified into classical MD and *ab-initio* MD, where ab-initio MD calculates the forces using a quantum-mechanics-based method, such as DFT, while classical MD uses an approximated function fit to ab-initio calculations to predict the atomic forces. More recently, the **machine learning methods for MD** have opened a new perspective by utilizing the group symmetric tools for geometric representation and the neural ODE Chen et al. (2018).

On one hand, many works have studied the protein-ligand binding problem in the equilibrium state (Stepniewska-Dziubinska et al., 2018; Jiménez et al., 2018; Jones et al., 2021; Yang et al., 2023), but not the MD simulation for binding dynamics. On the other hand, existing machine learning (ML) methods have studied molecular simulation (Zhang et al., 2018; Doerr et al., 2020; Musaelian et al., 2023; Fu et al., 2022b), but they are mainly studying small molecules or proteins. In this work, we propose an ML framework as an MD simulation to learn the protein-ligand binding dynamics.

## B   VISUAL ANALYSIS

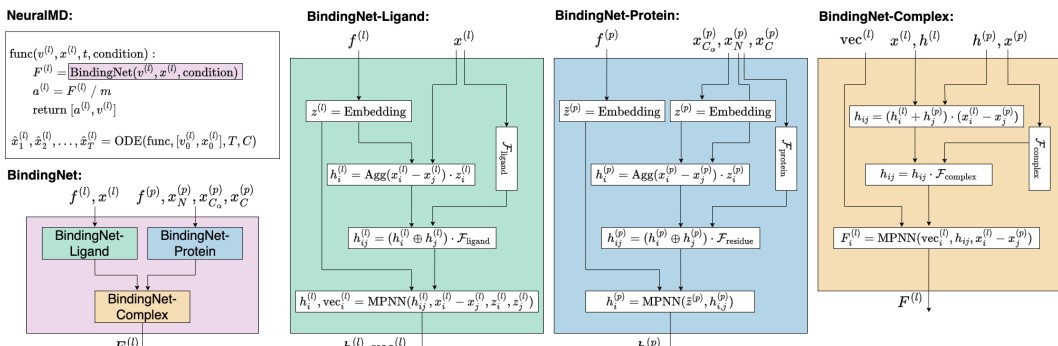

Figure 2: Brief pipeline of NeuralMD. In the three key modules of BindingNet, there are three vertical boxes, corresponding to three granularities of vector frames, as in Equations (23) to (25). More details are in Appendix F.

## C  GROUP SYMMETRY AND EQUIVARIANCE

In this article, a 3D molecular graph is represented by a collection of 3D point clouds. The corresponding symmetry group is SE(3), which consists of translations and rotations. Recall that we define the notion of equivariance functions in $\mathbf{R}^3$ in the main text through group actions. Formally, the group SE(3) is said to act on $\mathbf{R}^3$ if there is a mapping $\phi : \text{SE}(3) \times \mathbf{R}^3 \to \mathbf{R}^3$ satisfying the following two conditions:

1. if $e \in \text{SE}(3)$ is the identity element, then

$$\phi(e, \boldsymbol{r}) = \boldsymbol{r} \quad \text{for } \forall \boldsymbol{r} \in \mathbf{R}^3.$$

2. if $g_1, g_2 \in \text{SE}(3)$, then

$$\phi(g_1, \phi(g_2, \boldsymbol{r})) = \phi(g_1 g_2, \boldsymbol{r}) \quad \text{for } \forall \boldsymbol{r} \in \mathbf{R}^3.$$

Then, there is a natural SE(3) action on vectors $\boldsymbol{r}$ in $\mathbf{R}^3$ by translating $\boldsymbol{r}$ and rotating $\boldsymbol{r}$ for multiple times. For $g \in \text{SE}(3)$ and $\boldsymbol{r} \in \mathbf{R}^3$, we denote this action by $g\boldsymbol{r}$. Once the notion of group action is defined, we say a function $f : \mathbf{R}^3 \to \mathbf{R}^3$ that transforms $\boldsymbol{r} \in \mathbf{R}^3$ is equivariant if:

$$f(g\boldsymbol{r}) = gf(\boldsymbol{r}), \quad \text{for } \forall \ \boldsymbol{r} \in \mathbf{R}^3.$$

On the other hand, $f : \mathbf{R}^3 \to \mathbf{R}^1$ is invariant, if $f$ is independent of the group actions:

$$f(g\boldsymbol{r}) = f(\boldsymbol{r}), \quad \text{for } \forall \ \boldsymbol{r} \in \mathbf{R}^3.$$

For some scenarios, our problem is chiral sensitive. That is, after mirror reflecting a 3D molecule, the properties of the molecule may change dramatically. In these cases, it's crucial to include reflection transformations into consideration. More precisely, we say an SE(3) equivariant function $f$ is **reflection anti-symmetric**, if:

$$f(\rho \boldsymbol{r}) \neq f(\boldsymbol{r}), \tag{6}$$

for reflection $\rho \in \text{E}(3)$.

# D EQUIVARIANT VECTOR FRAMES

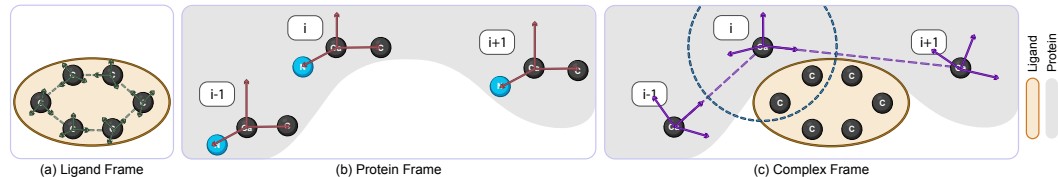

Figure 3: Three granularities of vector frame basis in BindingNet: (a) atom-level basis for ligands, (b) backbone-level basis for proteins, and (c) residue-level basis for the protein-ligand complex.

**Frame** is a popular terminology in science areas. In physics, the frame is equivalent to a coordinate system. For example, we may assign a frame to all observers, although different observers may collect different data under different frames, the underlying physics law should be the same. In other words, denote the physics law by $f$, then $f$ should be an equivariant function.

There are certain ways to choose the frame basis, and below we introduce two main types: the orthogonal basis and the protein backbone basis. The orthogonal basis can be built for flexible 3D point clouds such as atoms, while the protein backbone basis is specifically proposed to capture the protein backbone.

## D.1 BASIS

Since there are three orthogonal directions in $\mathbf{R}^3$, one natural frame in $\mathbf{R}^3$ can be a frame consisting of three orthogonal vectors:
$$F = (\boldsymbol{e}_1, \boldsymbol{e}_2, \boldsymbol{e}_3).$$
Once equipped with a frame (coordinate system), we can project all geometric quantities to this frame. For example, an abstract vector $\boldsymbol{x} \in \mathbf{R}^3$ can be written as $\boldsymbol{x} = (r_1, r_2, r_3)$ under the frame $F$, if: $\boldsymbol{x} = r_1\boldsymbol{e}_1 + r_2\boldsymbol{e}_2 + r_3\boldsymbol{e}_3$. An equivariant frame further requires the three orthonormal vectors in $(\boldsymbol{e}_1, \boldsymbol{e}_2, \boldsymbol{e}_3)$ to be equivariant. Intuitively, an equivariant frame will transform according to the global rotation or translation of the whole system. Once equipped with an equivariant frame, we can project equivariant vectors into this frame:

$$\boldsymbol{x} = \tilde{r}_1\boldsymbol{e}_1 + \tilde{r}_2\boldsymbol{e}_2 + \tilde{r}_3\boldsymbol{e}_3. \tag{7}$$

We call the process of $\boldsymbol{x} \to \tilde{r} := (\tilde{r}_1, \tilde{r}_2, \tilde{r}_3)$ the **scalarization** or **projection** operation. Since $\tilde{r}_i = \boldsymbol{e}_i \cdot \boldsymbol{x}$ is expressed as an inner product between equivariant vectors, we know that $\tilde{r}$ consists of scalars.

In this article, we assign an equivariant frame to each node/edge, therefore we call them the local frames. Given two atoms with 3D positions $(\boldsymbol{x}_i, \boldsymbol{x}_j)$, we can find the atom (denoted by $\boldsymbol{x}_k$) that is nearest to the center of $(\boldsymbol{x}_i, \boldsymbol{x}_j)$ by KNN algorithms. Then the equivariant frame is defined by:

$$\text{Vector-Frame}(\boldsymbol{x}_i, \boldsymbol{x}_j) := \textbf{Gram-Schmidt}\{\boldsymbol{x}_i - \boldsymbol{x}_j, \boldsymbol{x}_i - \boldsymbol{x}_k, (\boldsymbol{x}_i - \boldsymbol{x}_j) \times (\boldsymbol{x}_i - \boldsymbol{x}_k)\}. \tag{8}$$

The Gram-Schmidt orthogonalization makes sure that the Vector-Frame$(\boldsymbol{x}_i, \boldsymbol{x}_j)$ is orthonormal.

**Reflection Antisymmetric** Since we implement the cross product $\times$ for building the local frames, the third vector in the frame is a pseudo-vector. Then, the **projection** operation is not invariant under reflections (the inner product between a vector and a pseudo-vector change signs under reflection). Therefore, our model can discriminate two 3D geometries with different chiralities.

Our local frames also enable us to output equivariant vectors by multiplying scalars $(v_1, v_2, v_3)$ with the frame: $\boldsymbol{v} = v_1 \cdot \boldsymbol{e}_1 + v_2 \cdot \boldsymbol{e}_2 + v_3 \cdot \boldsymbol{e}_3$.

**Equivariance w.r.t. cross-product** The goal is to prove that the cross-product is equivariant to the SE(3)-group, *i.e.*:
$$gx \times gy = g(x \times y), \qquad g \in \text{SE(3)-Group} \tag{9}$$

**Geometric proof.** From intuition, with rotation matrix $g$, we are transforming the whole basis, thus the direction of $gx \times gy$ changes equivalently with $g$. And for the value/length of $gx \times gy$, because $|gx \times gy| = \|gx\| \cdot \|gy\| \cdot \sin\theta = \|x\| \cdot \|y\| \cdot \sin\theta = |x \times y|$. So the length stays the same, and the direction changes equivalently. Intuitively, this interpretation is quite straightforward.

**Analytical proof.** A more rigorous proof can be found below: First, we have that for the rotation matrix $g$:

$$
gx \times gy = \begin{bmatrix} \boldsymbol{g}_1^T \boldsymbol{x} \\ \boldsymbol{g}_2^T \boldsymbol{x} \\ \boldsymbol{g}_3^T \boldsymbol{x} \end{bmatrix} \times \begin{bmatrix} \boldsymbol{g}_1^T \boldsymbol{y} \\ \boldsymbol{g}_2^T \boldsymbol{y} \\ \boldsymbol{g}_3^T \boldsymbol{y} \end{bmatrix} = \begin{bmatrix} \boldsymbol{g}_2^T \boldsymbol{x} \cdot \boldsymbol{g}_3^T \boldsymbol{y} - \boldsymbol{g}_3^T x \cdot \boldsymbol{g}_2^T \boldsymbol{y} \\ -\boldsymbol{g}_1^T \boldsymbol{x} \cdot \boldsymbol{g}_3^T \boldsymbol{y} + \boldsymbol{g}_3^T \boldsymbol{x} \cdot \boldsymbol{g}_1^T \boldsymbol{y} \\ \boldsymbol{g}_1^T \boldsymbol{x} \cdot \boldsymbol{g}_2^T \boldsymbol{y} - \boldsymbol{g}_2^T \boldsymbol{x} \cdot \boldsymbol{g}_1^T \boldsymbol{y} \end{bmatrix},
\tag{10}
$$

where $\boldsymbol{g}_i, \boldsymbol{x}, \boldsymbol{y} \in \mathbb{R}^{3 \times 1}$.

Because $A^T C \cdot B^T D - A^T D \cdot B^T C = (A \times B)^T (C \times D)$, so we can have:

$$
gx \times gy = \begin{bmatrix} \boldsymbol{g}_2^T \boldsymbol{x} \cdot \boldsymbol{g}_3^T \boldsymbol{y} - \boldsymbol{g}_3^T x \cdot \boldsymbol{g}_2^T \boldsymbol{y} \\ -\boldsymbol{g}_1^T \boldsymbol{x} \cdot \boldsymbol{g}_3^T \boldsymbol{y} + \boldsymbol{g}_3^T \boldsymbol{x} \cdot \boldsymbol{g}_1^T \boldsymbol{y} \\ \boldsymbol{g}_1^T \boldsymbol{x} \cdot \boldsymbol{g}_2^T \boldsymbol{y} - \boldsymbol{g}_2^T \boldsymbol{x} \cdot \boldsymbol{g}_1^T \boldsymbol{y} \end{bmatrix} = \begin{bmatrix} (\boldsymbol{g}_2 \times \boldsymbol{g}_3)^T (\boldsymbol{x} \times \boldsymbol{y}) \\ (\boldsymbol{g}_3 \times \boldsymbol{g}_1)^T (\boldsymbol{x} \times \boldsymbol{y}) \\ (\boldsymbol{g}_1 \times \boldsymbol{g}_2)^T (\boldsymbol{x} \times \boldsymbol{y}). \end{bmatrix}
\tag{11}
$$

Then because:

$$
\det(g) = (\boldsymbol{g}_2 \times \boldsymbol{g}_3)^T \boldsymbol{g}_1 = \boldsymbol{g}_1^T \boldsymbol{g}_1 = 1
$$
$$
\Longrightarrow (\boldsymbol{g}_2 \times \boldsymbol{g}_3)^T \boldsymbol{g}_1 \boldsymbol{g}_1^{-1} = \boldsymbol{g}_1^T \boldsymbol{g}_1 \boldsymbol{g}_1^{-1}
$$
$$
\Longrightarrow (\boldsymbol{g}_2 \times \boldsymbol{g}_3)^T = \boldsymbol{g}_1^T.
\tag{12}
$$

Thus, we can have

$$
gx \times gy = \begin{bmatrix} (\boldsymbol{g}_2 \times \boldsymbol{g}_3)^T (\boldsymbol{x} \times \boldsymbol{y}) \\ (\boldsymbol{g}_3 \times \boldsymbol{g}_1)^T (\boldsymbol{x} \times \boldsymbol{y}) \\ (\boldsymbol{g}_1 \times \boldsymbol{g}_2)^T (\boldsymbol{x} \times \boldsymbol{y}) \end{bmatrix} = \begin{bmatrix} \boldsymbol{g}_1^T (\boldsymbol{x} \times \boldsymbol{y}) \\ \boldsymbol{g}_2^T (\boldsymbol{x} \times \boldsymbol{y}) \\ \boldsymbol{g}_3^T (\boldsymbol{x} \times \boldsymbol{y}) \end{bmatrix} = g(\boldsymbol{x} \times \boldsymbol{y}).
\tag{13}
$$

---

**Rotation symmetric** The goal is to prove

$$
\text{Vector-Frame}(g\boldsymbol{x}_i, g\boldsymbol{x}_j) = g\textbf{Gram-Schmidt}\{\boldsymbol{x}_i - \boldsymbol{x}_j, \boldsymbol{x}_i - \boldsymbol{x}_k, (\boldsymbol{x}_i - \boldsymbol{x}_j) \times (\boldsymbol{x}_i - \boldsymbol{x}_k)\}.
\tag{14}
$$

---

Thus we can have:

$$
\begin{aligned}
\text{Vector-Frame}(g\boldsymbol{x}_i, g\boldsymbol{x}_j) &= \textbf{Gram-Schmidt}\{g\boldsymbol{x}_i - g\boldsymbol{x}_j, g\boldsymbol{x}_i - g\boldsymbol{x}_k, (g\boldsymbol{x}_i - g\boldsymbol{x}_j) \times (g\boldsymbol{x}_i - g\boldsymbol{x}_k)\} \\
&= \textbf{Gram-Schmidt}\{g(\boldsymbol{x}_i - \boldsymbol{x}_j), g(\boldsymbol{x}_i - \boldsymbol{x}_k), g((\boldsymbol{x}_i - \boldsymbol{x}_j) \times (\boldsymbol{x}_i - \boldsymbol{x}_k))\}.
\end{aligned}
\tag{15}
$$

Recall that Gram-Schmidt projection (**Gram-Schmidt**$\{\boldsymbol{v}_1, \boldsymbol{v}_2, \boldsymbol{v}_3\}$) is:

$$
\begin{aligned}
\boldsymbol{u}_1 &= \boldsymbol{v}_1, & \boldsymbol{e}_1 &= \frac{\boldsymbol{v}_1}{\|\boldsymbol{v}_1\|}, \\
\boldsymbol{u}_2 &= \boldsymbol{v}_2 - \frac{\boldsymbol{u}_1^T \boldsymbol{v}_2}{\|\boldsymbol{u}_1\|} \boldsymbol{u}_1, & \boldsymbol{e}_2 &= \frac{\boldsymbol{v}_2}{\|\boldsymbol{v}_2\|}, \\
\boldsymbol{u}_3 &= \boldsymbol{v}_3 - \frac{\boldsymbol{u}_1^T \boldsymbol{v}_3}{\|\boldsymbol{u}_1\|} \boldsymbol{u}_1 - \frac{\boldsymbol{u}_2^T \boldsymbol{v}_3}{\|\boldsymbol{u}_2\|} \boldsymbol{u}_2, & \boldsymbol{e}_3 &= \frac{\boldsymbol{v}_3}{\|\boldsymbol{v}_3\|}.
\end{aligned}
\tag{16}
$$

Thus, the Gram-Schmidt projection on the rotated vector (**Gram-Schmidt**$\{g\boldsymbol{v}_1, g\boldsymbol{v}_2, g\boldsymbol{v}_3\}$) is:

$$
\begin{aligned}
\boldsymbol{u}_1' &= g\boldsymbol{v}_1, \\
\boldsymbol{u}_2' &= g\boldsymbol{v}_2 - g\frac{\boldsymbol{u}_1^T \boldsymbol{v}_2}{\|\boldsymbol{u}_1\|} \boldsymbol{u}_1, \\
\boldsymbol{u}_3' &= g\boldsymbol{v}_3 - g\frac{\boldsymbol{u}_1^T \boldsymbol{v}_3}{\|\boldsymbol{u}_1\|} \boldsymbol{u}_1 - g\frac{\boldsymbol{u}_2^T \boldsymbol{v}_3}{\|\boldsymbol{u}_2\|} \boldsymbol{u}_2,
\end{aligned}
\tag{17}
$$

Thus, **Gram-Schmidt**$\{g\boldsymbol{v}_1, g\boldsymbol{v}_2, g\boldsymbol{v}_3\} = g\textbf{Gram-Schmidt}\{\boldsymbol{v}_1, \boldsymbol{v}_2, \boldsymbol{v}_3\}$.

---

**Transition symmetric**

$$\text{Vector-Frame}(\boldsymbol{x}_i + \delta\boldsymbol{x}, \boldsymbol{x}_j + \delta\boldsymbol{x}) = \textbf{Gram-Schmidt}\{\boldsymbol{x}_i - \boldsymbol{x}_j, \boldsymbol{x}_i - \boldsymbol{x}_k, (\boldsymbol{x}_i - \boldsymbol{x}_j) \times (\boldsymbol{x}_i - \boldsymbol{x}_k)\}.$$

(18)

---

Because the basis is based on the difference of coordinates, it is straightforward to observe that **Gram-Schmidt**$\{\boldsymbol{v}_1 + \boldsymbol{t}, \boldsymbol{v}_2 + \boldsymbol{t}, \boldsymbol{v}_3 + \boldsymbol{t}\} = $ **Gram-Schmidt**$\{\boldsymbol{v}_1, \boldsymbol{v}_2, \boldsymbol{v}_3\}$. So the frame operation is transition equivariant.

---

**Reflection antisymmetric**

$$\text{Vector-Frame}(\boldsymbol{x}_i, \boldsymbol{x}_j) \neq \text{Vector-Frame}(-\boldsymbol{x}_i, -\boldsymbol{x}_j).$$

(19)

---

From intuition, this makes sense because the cross-product is anti-symmetric.

A simple counter-example is the original geometry $R$ and the reflected geometry by the original point $-R$. Thus the two bases before and after the reflection group is the following:

$$\textbf{Gram-Schmidt}\{\boldsymbol{x}_i - \boldsymbol{x}_j, \boldsymbol{x}_i - \boldsymbol{x}_k, (\boldsymbol{x}_i - \boldsymbol{x}_j) \times (\boldsymbol{x}_i - \boldsymbol{x}_k)\} \tag{20}$$

$$\textbf{Gram-Schmidt}\{-\boldsymbol{x}_i + \boldsymbol{x}_j, -\boldsymbol{x}_i + \boldsymbol{x}_k, (\boldsymbol{x}_i - \boldsymbol{x}_j) \times (\boldsymbol{x}_i - \boldsymbol{x}_k)\}. \tag{21}$$

The bases between $\boldsymbol{v}_1, \boldsymbol{v}_2, \boldsymbol{v}_3$ and $\{-\boldsymbol{v}_1, -\boldsymbol{v}_2, \boldsymbol{v}_3\}\}$ are different, thus such frame construction is reflection anti-symmetric.

### D.2 SCALARIZATION

Once we have the three vectors as the vector frame basis, the next thing is to do modeling. Suppose the frame is $\mathcal{F} = (\boldsymbol{e}_1, \boldsymbol{e}_2, \boldsymbol{e}_3)$, then for an equivariant vector (tensor) $\boldsymbol{h}$, the scalarization is:

$$\boldsymbol{h} \odot \mathcal{F} = (\boldsymbol{h} \odot \boldsymbol{e}_1, \boldsymbol{h} \odot \boldsymbol{e}_2, \boldsymbol{h} \odot \boldsymbol{e}_3) = (\boldsymbol{h}_1, \boldsymbol{h}_2, \boldsymbol{h}_3). \tag{22}$$

### D.3 MULTI-GRAINED SE(3)-EQUIVARIANT VECTOR FRAME

Proteins are essentially macromolecules composed of thousands of residues (amino acids), where each residue is a small molecule. Thus, it is infeasible to model all the atoms in proteins due to the large volume of the system, and such an issue also holds for the protein-ligand complex. To address this issue, we propose BindingNet, a multi-grained SE(3)-equivariant model. The vector frame basis ensures SE(3)-equivariance, and the multi-granularity is achieved by considering frames at three levels.

**Vector Frame Basis for SE(3)-Equivariance.** Recall that the geometric representation of the whole molecular system needs to follow the physical properties of the equivariance w.r.t. rotation and translation. Such a group symmetric property is called SE(3)-equivariance. We also want to point out that the reflection or chirality property is equivariant for properties like energy, yet it is not for the ligand modeling with rigid protein structures (*i.e.*, antisymmetric to the reflection). The vector frame basis can handle this naturally, and we leave a more detailed discussion in Appendix D, along with the proof on group symmetry of vector frame basis. In the following, we introduce three levels of vector frames for multi-grained modeling.

**Atom-Level Vector Frame for Ligands.** For small molecule ligands, we first extract atom pairs $(i, j)$ within the distance cutoff $c$, and the vector frame basis is constructed using the Gram-Schmidt as:

$$\mathcal{F}_{\text{ligand}} = \left(\frac{\boldsymbol{x}_i^{(l)} - \boldsymbol{x}_j^{(l)}}{\left\|\boldsymbol{x}_i^{(l)} - \boldsymbol{x}_j^{(l)}\right\|}, \frac{\boldsymbol{x}_i^{(l)} \times \boldsymbol{x}_j^{(l)}}{\left\|\boldsymbol{x}_i^{(l)} \times \boldsymbol{x}_j^{(l)}\right\|}, \frac{\boldsymbol{x}_i^{(l)} - \boldsymbol{x}_j^{(l)}}{\left\|\boldsymbol{x}_i^{(l)} - \boldsymbol{x}_j^{(l)}\right\|} \times \frac{\boldsymbol{x}_i^{(l)} \times \boldsymbol{x}_j^{(l)}}{\left\|\boldsymbol{x}_i^{(l)} \times \boldsymbol{x}_j^{(l)}\right\|}\right), \tag{23}$$

where $\times$ is the cross product. Note that both $\boldsymbol{x}_i^{(l)}$ and $\boldsymbol{x}_j^{(l)}$ are for geometries at time $t$ - henceforth, we omit the subscript $t$ for brevity. Such an atom-level vector frame allows us to do SE(3)-equivariant message passing to get the atom-level representation.

**Backbone-Level Vector Frame for Proteins.** Proteins can be treated as chains of residues, where each residue possesses a backbone structure. The backbone structure comprises an amino group, a

carboxyl group, and an alpha carbon, delegated as $N - C_\alpha - C$. Such a structure serves as a natural way to build the vector frame. For each residue in the protein, the coordinates are $\boldsymbol{x}_N$, $\boldsymbol{x}_{C_\alpha}$, and $\boldsymbol{x}_C$, then the backbone-level vector frame for this residue is:

$$\mathcal{F}_{\text{protein}} = (\frac{\boldsymbol{x}_N - \boldsymbol{x}_{C_\alpha}}{\|\boldsymbol{x}_N - \boldsymbol{x}_{C_\alpha}\|}, \frac{\boldsymbol{x}_{C_\alpha} - \boldsymbol{x}_C}{\|\boldsymbol{x}_{C_\alpha} - \boldsymbol{x}_C\|}, \frac{\boldsymbol{x}_N - \boldsymbol{x}_{C_\alpha}}{\|\boldsymbol{x}_N - \boldsymbol{x}_{C_\alpha}\|} \times \frac{\boldsymbol{x}_{C_\alpha} - \boldsymbol{x}_C}{\|\boldsymbol{x}_{C_\alpha} - \boldsymbol{x}_C\|}). \tag{24}$$

This is built for each residue, providing a residue-level representation.

**Residue-Level Vector Frame for Protein-Ligand Complexes.** It is essential to model the protein-ligand interaction to better capture the binding dynamics. We achieve this by introducing the residue-level vector frame. More concretely, proteins are sequences of residues, marked as $\{(f_0^{(p)}, \boldsymbol{x}_0^{(p)}), ..., (f_i^{(p)}, \boldsymbol{x}_i^{(p)}), (f_{i+1}^{(p)}, \boldsymbol{x}_{i+1}^{(p)}, ...\}$. Here, we use a cutoff threshold $c$ to determine the interactions between ligands and proteins, and the interactive regions on proteins are called pockets. We construct the following vector frame for residues in the pockets sequentially:

$$\mathcal{F}_{\text{complex}} = (\frac{\boldsymbol{x}_i^{(p)} - \boldsymbol{x}_{i+1}^{(p)}}{\left\|\boldsymbol{x}_i^{(p)} - \boldsymbol{x}_{i+1}^{(p)}\right\|}, \frac{\boldsymbol{x}_i^{(p)} \times \boldsymbol{x}_{i+1}^{(p)}}{\left\|\boldsymbol{x}_i^{(p)} \times \boldsymbol{x}_{i+1}^{(p)}\right\|}, \frac{\boldsymbol{x}_i^{(p)} - \boldsymbol{x}_{i+1}^{(p)}}{\left\|\boldsymbol{x}_i^{(p)} - \boldsymbol{x}_{i+1}^{(p)}\right\|} \times \frac{\boldsymbol{x}_i^{(p)} \times \boldsymbol{x}_{i+1}^{(p)}}{\left\|\boldsymbol{x}_i^{(p)} \times \boldsymbol{x}_{i+1}^{(p)}\right\|}). \tag{25}$$

Through this complex-level vector frame, the message passing enables the exchange of information between atoms from ligands and residues from the pockets. The illustration of the above three levels of vector frames can be found in Figure 3. Once we build up such three vector frames, we then conduct a *scalarization* operation (Du et al., 2022), which transforms the equivariant variables (*e.g.*, coordinates) to invariant variables by projecting them to the three vector bases in the vector frame.

# E   SPECIFICATIONS ON MISATO

In this section, we provide more details on the MISATO dataset (Siebenmorgen et al., 2023). For small molecule ligands, we ignore the Hydrogen atoms.

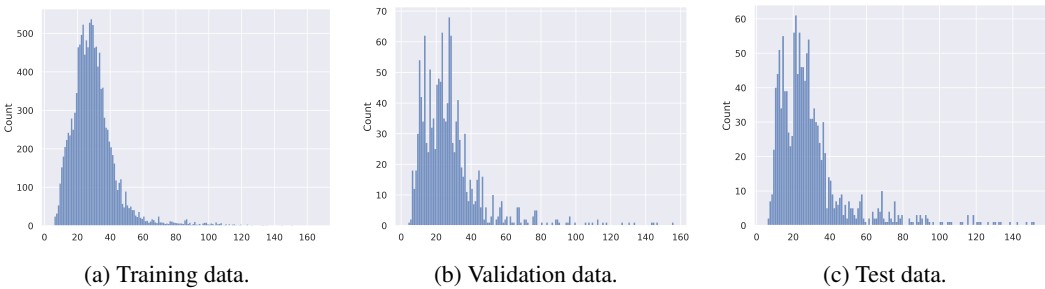

(a) Training data.                    (b) Validation data.                    (c) Test data.

Figure 4: Distribution on # atoms in small molecule ligands for all protein-ligand complex.

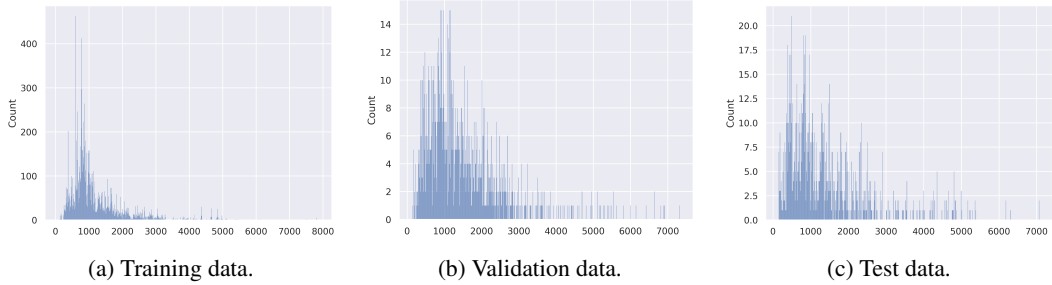

(a) Training data.                    (b) Validation data.                    (c) Test data.

Figure 5: Distribution on # residues in proteins for all protein-ligand complex.

We also plot the distribution of the energy gap between each time step and the initial snapshot, *i.e.*, $E_t - E_0$. The distribution is in Figure 6. We can observe that as the time processes, the mean of the energy stays almost the same, yet the variance gets higher.

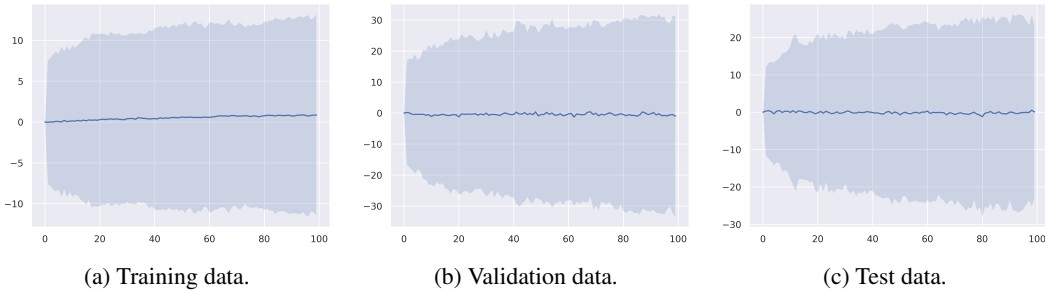

(a) Training data.                    (b) Validation data.                    (c) Test data.

Figure 6: Distribution on energy $E_t - E_0$.

# F DETAILS OF NEURALMD

**Residue-Level Complex Modeling.** Once we obtain the atom-level representation and vector $(\boldsymbol{h}^{(l)}, \text{vec}^{(l)})$ from ligands, and backbone-level representation $(\boldsymbol{h}^{(p)})$ from proteins, the next step is to learn the protein-ligand interaction. We first extract the residue-atom pair $(i, j)$ with a cutoff $c$, based on which we obtain an equivariant interaction edge representation $\boldsymbol{h}_{ij} = (\boldsymbol{h}_i^{(l)} + \boldsymbol{h}_j^{(p)}) \cdot (\boldsymbol{x}_i^{(l)} - \boldsymbol{x}_j^{(p)})$. After scalarization, we can obtain invariant interaction edge representation $\boldsymbol{h}_{ij} = \boldsymbol{h}_{ij} \cdot \mathcal{F}_{\text{complex}}$. Finally, we adopt an equivariant MPNN layer to get the atom-level force as:

$$\text{vec}_{ij}^{(pl)} = \text{vec}_i^{(l)} \cdot \text{MLP}(h_{ij}) + (x_i^{(l)} - x_j^{(p)}) \cdot \text{MLP}(h_{ij}), \quad F_i^{(l)} = \text{vec}_i^{(l)} + \text{Agg}_{j \in \mathcal{N}(i)} \text{vec}_{ij}^{(pl)}. \quad (26)$$

In the last equation, the ultimate force predictions can be viewed as two parts: the internal force from the molecule $\text{vec}_i^{(l)}$ and the external force from the protein-ligand interaction $\text{vec}_{ij}^{(pl)}$.

## F.1 ARCHITECTURE DETAILS

In this section, we provide more details on the model architecture in Figure 7, and hyperparameter details in Table 1.

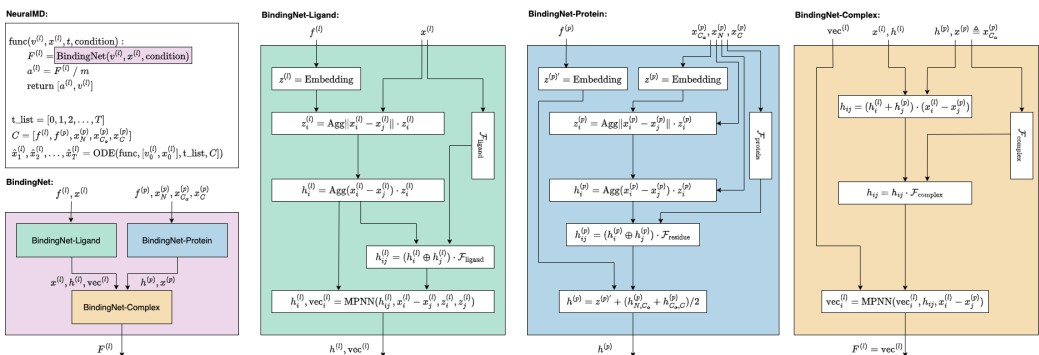

Figure 7: Detailed pipeline of NeuralMD. In the three key modules of BindingNet, there are three vertical boxes, corresponding to three granularities of vector frames, as in Equations (23) to (25).

Table 1: Hyperparameter specifications for NeuralMD.

| Hyperparameter | Value |
| --- | --- |
| # layers | {5} |
| $c$ | {5} |
| cutoff $c$ | 5 |
| learning rate | {1e-4, 1e-3} |
| optimizer | {SGD, Adam } |

Table 2: Results on ten single-trajectory binding dynamics prediction. Results with optimal training loss are reported. Four evaluation metrics are considered: MAE (Å, ↓), MSE (↓), and Stability (%, ↓).

| PDB ID | Metric | VerletMD | GNN-MD | DenoisingLD | NeuralMD ODE (Ours) | NeuralMD SDE (Ours) | PDB ID | Metric | VerletMD | GNN-MD | DenoisingLD | NeuralMD ODE (Ours) | NeuralMD SDE (Ours) |
|---|---|---|---|---|---|---|---|---|---|---|---|---|---|
| 5WIJ | MAE | 9.618 | 2.319 | 2.254 | 2.118 | **2.109** | 1XP6 | MAE | 13.444 | 2.303 | 1.915 | **1.778** | 1.822 |
| | MSE | 6.401 | 1.553 | 1.502 | 1.410 | **1.408** | | MSE | 9.559 | 1.505 | 1.282 | **1.182** | 1.216 |
| | Stability | 79.334 | 45.369 | 18.054 | **12.654** | 13.340 | | Stability | 86.393 | 43.019 | 28.417 | **19.256** | 22.734 |
| 4ZX0 | MAE | 21.033 | 2.255 | 1.998 | **1.862** | 1.874 | 4YUR | MAE | 15.674 | 7.030 | 6.872 | **6.807** | 6.826 |
| | MSE | 14.109 | 1.520 | 1.347 | **1.260** | 1.271 | | MSE | 10.451 | 4.662 | 4.520 | **4.508** | 4.526 |
| | Stability | 76.878 | 41.332 | 23.267 | **18.189** | 18.845 | | Stability | 81.309 | 50.238 | 32.423 | **23.250** | 25.008 |
| 3EOV | MAE | 25.403 | 3.383 | 3.505 | 3.287 | **3.282** | 4G3E | MAE | 5.181 | 2.672 | 2.577 | 2.548 | **2.478** |
| | MSE | 17.628 | 2.332 | 2.436 | 2.297 | **2.294** | | MSE | 3.475 | 1.743 | 1.677 | 1.655 | **1.615** |
| | Stability | 91.129 | 57.363 | 51.590 | **44.775** | 44.800 | | Stability | 65.377 | 16.365 | 7.188 | **2.113** | 2.318 |
| 4K6W | MAE | 14.682 | 3.674 | 3.555 | 3.503 | **3.429** | 6B7F | MAE | 31.375 | 4.129 | 3.952 | 3.717 | **3.657** |
| | MSE | 9.887 | 2.394 | 2.324 | 2.289 | **2.234** | | MSE | 21.920 | 2.759 | 2.676 | 2.503 | **2.469** |
| | Stability | 87.147 | 57.852 | 39.580 | 38.562 | **38.476** | | Stability | 87.550 | 54.900 | 16.050 | **3.625** | 22.750 |
| 1KTI | MAE | 18.067 | **6.534** | 6.657 | 6.548 | 6.537 | 3B9S | MAE | 19.347 | 2.701 | 2.464 | **2.351** | 2.374 |
| | MSE | 12.582 | 4.093 | 4.159 | 4.087 | **4.085** | | MSE | 11.672 | 1.802 | 1.588 | **1.527** | 1.542 |
| | Stability | 77.315 | 4.691 | 7.377 | 0.525 | **0.463** | | Stability | 41.667 | 43.889 | 8.819 | **0.000** | 0.000 |

# G   MORE EXPERIMENT RESULTS

## G.1   EXPERIMENT SETUP

To verify the effectiveness and efficiency of NeuralMD, we design ten single-trajectory and three multi-trajectory binding simulation tasks. For evaluation, we adopt the recovery and stability metrics (Fu et al., 2022a). NeuralMD achieves 2000× speedup compared to the numerical methods. We observe that NeuralMD outperforms all other ML methods (Zhang et al., 2018; Musaelian et al., 2023; Fu et al., 2022b; Wu & Li, 2023; Arts et al., 2023) on 12 tasks using recovery metric, and NeuralMD is consistently better by a large gap using the stability metric (up to ~80%). Qualitatively, we illustrate that NeuralMD realizes more stable binding dynamics predictions in three case studies. They are three protein-ligand binding complexes from Protein Data Bank (PDB), as shown in Figure 1.

**Baselines.** In this work, we mainly focus on machine learning methods for trajectory prediction, *i.e.*, no energy or force labels are considered. GNN-MD is to apply geometric graph neural networks (GNNs) to predict the trajectories in an auto-regressive manner (Siebenmorgen et al., 2023; Fu et al., 2022b). More concretely, GNN-MD takes as inputs the geometries at time $t$ and predicts the geometries at time $t + 1$. DenoisingLD (denoising diffusion for Langevin dynamics) (Arts et al., 2023; Wu & Li, 2023; Fu et al., 2022b) is a baseline method that models the trajectory prediction as denoising diffusion task (Song et al., 2020), and the inference for trajectory generation essentially becomes the Langevin dynamics. CG-MD learns a dynamic GNN and a score GNN (Fu et al., 2022b), which are essentially the hybrid of GNN-MD and DenoisingLD. Here, to make the comparison more explicit, we compare these two methods (GNN-MD and DenoisingLD) separately. Additionally, we consider VerletMD, an energy prediction research line (including DeePMD (Zhang et al., 2018), TorchMD (Doerr et al., 2020), and Allegro-LAMMPS (Musaelian et al., 2023)), where the role of ML models is to predict the energy, and the MD trajectory is obtained by the velocity Verlet algorithm, a numerical integration method for Newtonian mechanics. We keep the same backbone model (BindingNet) for energy or force prediction for all the baselines.

## G.2   MAIN RESULTS ON SINGLE-TRAJECTORY PREDICTION

The main results in shown in Table 2.

## G.3   MORE RESULTS ON SINGLE-TRAJECTORY PREDICTION

One main benefit of using NeuralMD for binding simulation is its efficiency. To show this, we list the computational time in Table 3. We further approximate the wall time of the numerical method for MD simulation (PDB 5WIJ). Concretely, we can get an estimated speed of 1 nanosecond of dynamics every 0.28 hours. This is running the simulation with GROMACS (Van Der Spoel et al., 2005) on 1 GPU with 16 CPU cores and a moderately sized water box at the all-atom level (with 2 femtosecond timesteps). This equivalently shows that NeuralMD is ~2000× faster than numerical methods.

Table 4: Results on three multi-trajectory binding dynamics predictions. Results with optimal validation loss are reported. Four evaluation metrics are considered: MAE (Å, ↓), MSE (↓), and Stability (%, ↓).

| Dataset | MISATO-100 | | | MISATO-1000 | | | MISATO-All | | |
|---|---|---|---|---|---|---|---|---|---|
| | MAE | MSE | Stability | MAE | MSE | Stability | MAE | MSE | Stability |
| VerletMD | 90.326 | 56.913 | 86.642 | 80.187 | 53.110 | 86.702 | 105.979 | 69.987 | 90.665 |
| GNN-MD | 7.176 | 4.726 | 35.431 | 7.787 | 5.118 | 33.926 | 8.260 | 5.456 | 32.638 |
| DenoisingLD | 7.112 | 4.684 | 29.956 | 7.746 | 5.090 | 18.898 | 15.878 | 10.544 | 89.586 |
| NeuralMD-ODE (Ours) | **6.852** | **4.503** | **19.173** | **7.653** | **5.028** | **15.572** | **8.147** | **5.386** | **17.468** |
| NeuralMD-SDE (Ours) | 6.869 | 4.514 | 19.561 | 7.665 | 5.037 | 16.501 | 8.165 | 5.398 | 19.012 |

Table 3: Efficiency comparison of FPS between VerletMD and NeuralMD on single-trajectory prediction.

| PDB ID | 5WIJ | 4ZX0 | 3EOV | 4K6W | 1KTI | 1XP6 | 4YUR | 4G3E | 6B7F | 3B9S | Average |
|---|---|---|---|---|---|---|---|---|---|---|---|
| VerletMD | 12.564 | 30.320 | 29.890 | 26.011 | 19.812 | 28.023 | 31.513 | 29.557 | 19.442 | 31.182 | 25.831 |
| NeuralMD (Ours) | 33.164 | 39.415 | 31.720 | 31.909 | 24.566 | 37.135 | 39.365 | 39.172 | 20.320 | 37.202 | 33.397 |

## G.4 MD PREDICTION: GENERALIZATION AMONG MULTIPLE TRAJECTORIES

A more challenging task is to test the generalization ability of NeuralMD among different trajectories. The MISATO dataset includes 13,765 protein-ligand complexes, and we first create two small datasets by randomly sampling 100 and 1k complexes, respectively. Then, we take 80%-10%-10% for training, validation, and testing. We also consider the whole MISATO dataset, where the data split has already been provided. After removing the peptide ligands, we have 13,066, 1,357, and 1,357 complexes for training, validation, and testing, respectively.

The quantitative results are in Table 4. First, we can observe that VerletMD has worse performance on all three datasets, and the performance gap with other methods is even larger compared to the single-trajectory prediction. The other two baselines, GNN-MD and DenoisingLD, show similar performance, while NeuralMD outperforms in all datasets. Notice that stability (%) remains more distinguishable than the two trajectory recovery metrics (MAE and MSE).

## G.5 ABLATION STUDIES: FLEXIBLE BINDING

Recall that, in the main paper, we have discussed using the *semi-flexible* binding setting, *i.e.*, proteins with rigid structures while small molecule ligands with flexible structures, and the goal is to predict the trajectories of the ligands. If we want to take both proteins and ligands with flexible structures, one limitation is the GPU memory cost. However, we would like to mention that it is possible to do NeuralMD on protein-ligand with small volume, and we take an ablation study to test them as below.

**Problem Setup.** Both the proteins and ligands are flexible, and we want to predict their trajectories simultaneously. In the main paper, we consider three levels of vector frames. Here in the flexible setting, due to the large volume of atoms in the protein-ligand complex, we are only able to consider two levels, *i.e.*, the atom-level and residue-level. Thus, the backbone model (BindingNet) also changes accordingly. The performance is shown in Table 5, and we can see that NeuralMD is consistently better than the GNN-MD on all three metrics and all 10 single trajectories. We omit the multi-trajectory experiments due to the memory limitation.

Table 5: Results on ten single-trajectory binding dynamics prediction. Results with optimal training loss are reported. Four evaluation metrics are considered: MAE (Å, ↓), MSE (↓), and Stability (%, ↓).

|  |  | GNN-MD | NeuralMD (Ours) |
|---|---|---|---|
| 5WIJ | MAE | 7.126 | **3.101** |
|  | MSE | 4.992 | **2.070** |
|  | Stability | 68.317 | **30.655** |
| 4ZX0 | MAE | 9.419 | **2.580** |
|  | MSE | 6.269 | **1.724** |
|  | Stability | 67.492 | **29.013** |
| 3EOV | MAE | 10.695 | **3.664** |
|  | MSE | 7.447 | **2.521** |
|  | Stability | 67.782 | **39.714** |
| 4K6W | MAE | 8.347 | **3.056** |
|  | MSE | 5.605 | **2.007** |
|  | Stability | 63.839 | **36.972** |
| 1KTI | MAE | 8.900 | **6.815** |
|  | MSE | 5.820 | **4.268** |
|  | Stability | 65.010 | **26.805** |
| 1XP6 | MAE | 8.496 | **1.910** |
|  | MSE | 5.673 | **1.276** |
|  | Stability | 70.019 | **33.907** |
| 4YUR | MAE | 11.710 | **7.568** |
|  | MSE | 7.759 | **4.966** |
|  | Stability | 69.163 | **34.636** |
| 4G3E | MAE | 1314.425 | **3.282** |
|  | MSE | 814.641 | **2.152** |
|  | Stability | 65.703 | **21.095** |
| 6B7F | MAE | 182.278 | **3.166** |
|  | MSE | 115.688 | **2.121** |
|  | Stability | 72.027 | **26.931** |
| 3B9S | MAE | 3.590 | **2.477** |
|  | MSE | 2.431 | **1.615** |
|  | Stability | 54.890 | **18.817** |

