# OpenReview forum: "A Multi-Grained Symmetric Differential Equation Model for Learning Protein-Ligand Binding Dynamics"
_ICLR.cc/2024/Workshop/AI4DiffEqtnsInSci — AI4DiffEqtnsInSci @ ICLR 2024 Oral_

### Official Review · Reviewer_eFrF · 2024-02-23
**Review for NeuralMD**

**Rating:** 8
**Confidence:** 3

**Review:**

The authors have introduced a ML surrogate model for learning the Protein-ligand binding dynamics. The manuscript is well written with all the relevant details explicitly mentioned.

I have few overall concerns in this topic, please address them below

1. ML surrogates are highly dependent on the dataset they are trained for. What exactly is the motivation here to develop ML surrogates here ?
2. Is there any real time computation of MD simulation that is really needed for warranting the development of ML surrogates
2. ML surrogates are not extrapolative in nature and perform poorly on cases outside the dataset considered for train and testing. In such as case how do we account for such outliers in online computation.

---

### Official Review · Reviewer_QhjU · 2024-02-26
**Authors propose NeuralMD which is a ML surrogate model, multi-grained symmetric differential equation model for learning protein-ligand binding dynamics.**

**Rating:** 10
**Confidence:** 5

**Review:**

Authors propose NeuralMD which is a ML surrogate model, multi-grained symmetric differential equation model for learning protein-ligand binding dynamics.

---

### Meta-Review · Program_Chairs · 2024-03-03

**Recommendation:** Accept (Oral)

**Metareview:**

This is a high-quality paper. Both reviewers mark this as a high accept with a strong application area. I would recommend this paper as an oral.

---

### Decision · Program_Chairs · 2024-03-03

Accept (Oral)